# Are we really Bayesian? Probabilistic inference shows sub-optimal knowledge transfer

**Chin-Hsuan Sophie Lin**[1]*, **Trang Thuy Do**[1], **Lee Unsworth**[1], **Marta I. Garrido**[1,2]

1 Melbourne School of Psychological Sciences, The University of Melbourne, Melbourne, Australia,
2 Graeme Clark Institute for Biomedical Engineering, The University of Melbourne, Melbourne, Australia

* chinhsuan.lin@unimelb.edu.au

## Abstract

Numerous studies have found that the Bayesian framework, which formulates the optimal integration of the knowledge of the world (i.e. prior) and current sensory evidence (i.e. likelihood), captures human behaviours sufficiently well. However, there are debates regarding whether humans use precise but cognitively demanding Bayesian computations for behaviours. Across two studies, we trained participants to estimate hidden locations of a target drawn from priors with different levels of uncertainty. In each trial, scattered dots provided noisy likelihood information about the target location. Participants showed that they learned the priors and combined prior and likelihood information to infer target locations in a Bayes fashion. We then introduced a transfer condition presenting a trained prior and a likelihood that has never been put together during training. How well participants integrate this novel likelihood with their learned prior is an indicator of whether participants perform Bayesian computations. In one study, participants experienced the newly introduced likelihood, which was paired with a different prior, during training. Participants changed likelihood weighting following expected directions although the degrees of change were significantly lower than Bayes-optimal predictions. In another group, the novel likelihoods were never used during training. We found people integrated a new likelihood within (interpolation) better than the one outside (extrapolation) the range of their previous learning experience and they were quantitatively Bayes-suboptimal in both. We replicated the findings of both studies in a validation dataset. Our results showed that Bayesian behaviours may not always be achieved by a full Bayesian computation. Future studies can apply our approach to different tasks to enhance the understanding of decision-making mechanisms.

## Author summary

Bayesian decision theory has emerged as a unified approach for capturing a wide range of behaviours under uncertainty. However, behavioural evidence supporting that humans use explicit Bayesian computation is scarce. While it has been argued that knowledge generalization should be treated as hard evidence of the use of Bayesian strategies, results from previous work were inconclusive. Here, we develop a marker that effectively quantifies how well humans transfer learned priors to a new scenario. Our marker can be applied

**Data Availability Statement:** The experimental data and Matlab code to reproduce the main figures and to fit the models will be available at https://github.com/sophietwlim/BayesTransferProj

**Funding:** CSL and MIG were funded by the Australian Research Council Centre of Excellence for Integrative Brain Function (ARC Centre Grant CE140100007). CSL received a salary from the Australian Research Council Centre of Excellence for Integrative Brain Function (ARC Centre Grant CE140100007), which was awarded to MIG. CSL was funded by the University of Melbourne for Early Career Research Grant (2021ECR104). The funders had no role in study design, data collection and analysis, decision to publish, or preparation of the manuscript. URL of the Australian Research Council: https://www.arc.gov.au/ URL of the University of Melbourne: https://www.unimelb.edu.au/.

**Competing interests:** The authors have declared that no competing interests exist.

to various tasks and thus can provide a path forwarding the understanding of psychological and biological underpinnings of inferential behaviours.

## Introduction

How should we make sensible decisions using uncertain and ambiguous information? This is a major challenge we face in daily life. Bayesian decision theory (BDT) posits that probabilistically rational decisions can be reached by integrating knowledge about the environment (i.e. prior) together with current sensory inputs (i.e. likelihood) based on their respective reliabilities [1]. Some evidence has shown that across various perceptual and cognitive tasks [2–8] behaviours are close to Bayes-optimal. However, there are behaviours which are qualitatively Bayesian (i.e., following the reliability-based weighting principle) but fall short of Bayes-optimality [9], hence casting doubt upon the ideal Bayesian-observer theory. Studies have also shown cases where neurotypical [10] or neurodivergent populations [11,12] can fail to represent the true probability distribution of the world. These findings brought into questions whether the brain computes behaviours using the Bayes rule [13–15]. It has been argued that given the complexity of Bayesian computations, the brain may well use simpler approximations to achieve Bayes fashioned behaviours [15,16]. Understanding the exact strategies behind decisions under uncertainty is important given how ubiquitous uncertainty is in every real-life decision [17].

We have previously [18] reviewed behaviours that fell short of Bayesian optimality but can still be better explained by Bayesian than heuristic strategies. We argued that examining how well performance matches Bayes-optimum is not effective in resolving whether computation conforms to Bayesian expectations, let alone understanding how the brain makes probabilistic decision. Maloney and Mamassian [19] have proposed using a "generalization" criterion instead to evaluate whether the brain indeed makes inference in a Bayesian way. While people can learn a Bayes optimality policy by trial and error, these kinds of rote learners need to re-learn the policy whenever a change occurs in a learned task. As a result, they would likely fall short of the mark right after the change. On the contrary, a truly Bayesian agent that fully learns and flexibly accesses the components of the BDT will generalize efficiently by integrating new changes in the likelihood to any known priors [20]. Generalization can be experimentally confirmed if a learned prior is transferred to novel conditions. This process has been coined as *Bayesian transfer* [19]. We need to stress that *transfer* itself still concerns the computation strategies, meaning that it does not answer which algorithm the brain uses to approximate Bayesian computation. However, the pattern of a divergence from optimal transfer can inform not only computation strategies but also guide our search for corresponding algorithms that suitably produce such strategies. For example, some studies have theorised how some algorithms implemented in the brain can compute the underlying statistical information of the world [21]. These theories provide predictions about the patterns of generalization to unseen data beyond the context of past observations. There are also theories of Bayesian approximators that predict "generalization" behaviours which could only be observed within the context of past observations [22,23]. We can see how examining transfer is thus more informative than optimality matching in advancing our understanding of how the brain makes probabilistic decision-making.

A few studies have investigated Bayesian transfer in perception. In tasks where strong priors from years of experience are used (e.g. inferring object locations given auditory and visual cues such as locating a vehicle from its image and engine sounds in daily life; ref [24]) or when

prior probability distributions are visually presented [25,26], people instantly integrate changes of prior-likelihood pairs, which has been interpreted as successful transfer. However, where two embedded location priors were acquired through trial-by-trial feedback during a task [27], participants failed to show a robust transfer effect when a new likelihood was introduced. The authors of this study [27] concluded that in their task, the complexity of two priors and the cognitive demand of acquiring them simultaneously may have prompted the use of a computationally less expensive non-Bayesian strategy, and hence a weak transfer. These studies suggested that the cognitive load of learning and maintaining priors might be a deciding factor on whether people use a Bayesian strategy.

Besides prior complexity, the characteristics of sensory inputs (likelihoods) encountered during transfer also affect transfer effectiveness. Studies observed successful transfer when a likelihood distribution has either been acquired through lifelong experience [24,28] (in both studies the manipulated sensory reliability is visual contrast) or used during learning [25,26] (in both studies the manipulated sensory reliability is how likely a cue location is the true target location and all the cues used in the transfer tasks were presented during learning albeit with a prior not used in the transfer). Conversely, Kiryakova and colleagues [27] evaluated transfer by introducing a likelihood that participants had never experienced when they learned the location task and found trivial transfer. This difference touches on a fundamental issue about inferential behaviours. It is often considered that the human mind excels at applying knowledge to unseen data and untrained tasks [29,30]. Many argued that structured world knowledge which can represent the generative processes of physical stimuli is needed to enable such powerful generalization. The ability of the Bayesian model to explain how humans build abstract knowledge from sensory inputs is thus strongly appealing to neuroscientists [31,32]. However, an alternative hypothesis is that the seemingly powerful generalization is an illusion caused by the hardware (i.e. our brains) that has long evolved to fit the world we live in well [29,33] and been optimised with extensive training data encountered during lifetime [23]. Based on this hypothesis, nearly perfect generalization in response to novel likelihoods only occur when these likelihoods are physical features that are either well represented in the brain as a result of natural selection (such as luminance, ref [19]), or within the context of past experience [22]. On the other hand, generalizing outside the natural selection process or living experience would be suboptimal. Overall results from the abovementioned transfer studies seem to support predictions by this "bounded optimal" hypothesis. However, as have been described, the weak transfer in [27] could have simply resulted from multiple prior learning so more studies with proper designs that can discriminate effects from each factor are needed.

"Successful transfer" has been defined by behaviours being close to Bayes-optimal in novel conditions according to objective prior/likelihood distributions. However, research has repeatedly shown that humans can behave qualitatively Bayesian and yet represent prior/likelihood uncertainty differently from objective parameters [9,26,34]. Importantly, in these studies, Bayesian models still explain behaviours better than non-Bayesian models, meaning that neither can imperfect prior/likelihood weighting exclude the use of Bayesian strategy, nor can close-to-Bayes-optimal be taken as evidence for transfer. In fact, Kiryakova and colleagues [27] acknowledged that a lack of transfer in their study may be due to a small effect size caused by biased uncertainty estimation. Therefore, examining transfer using a proper mathematically operationalised criterion that can separate the influence of imperfect prior/likelihood estimation from suboptimal transfer is needed before we can draw any conclusions about the effect of cognitive demand on the use of Bayesian strategies or consider any implications of transferring outside training data.

In this study, we propose a mathematical definition of transfer that is prior/likelihood independent and provides a quantification of knowledge generalization. We applied a visual-spatial

task (**Fig 1A and 1B**) adapted from [35] in which the location of a hidden target is sampled from prior distributions (with low or high variance that participants learn from location feedback) and scattered dots provide noisy likelihood information about target location each trial (with variance manipulated using dot dispersion). Human participants could use both prior and likelihood to estimate positions of hidden visual targets. We report two experiments. The first asked whether the use of Bayesian strategies depends on cognitive loads. To do so, we compared transfer performance between sequential and simultaneous learning of two priors (**Fig 1C**). Based on past studies, we hypothesised that transfer would be worse in simultaneous learning due to its higher cognitive demand. The second experiment asked whether knowledge transfer depends on generalizing within (i.e., interpolation) or outside learning (i.e., extrapolation) conditions (**Fig 1D**). Based on past studies we hypothesised that transfer would be close to optimal in the interpolation but suboptimal in the extrapolation condition. Further, in response to the replication crisis [36], for both experiments, we present data from a discovery and an independent validation dataset.

## Results

**Experiment 1.** Linear mixed models were used for (**Fig 3A**, **3B and 3D**) for data which were not normally distributed. We used likelihood ratio tests to compare linear mixed-effect models that were designed to delineate the effects of prior, likelihood, and learning group. ANOVA and t-test were used for the transfer score of the discovery set (**Fig 3C**).

## Learning phase

Discovery set A model with the prior and likelihood fixed effects explained slope data best. We showed that participants were qualitatively Bayesian (**Fig 3A**), i.e. weighting according to the reliability of prior (linear mixed model $p = 5.43 \text{ X } 10^{-13}$) and likelihood (linear mixed model $p = 1.57 \text{ X } 10^{-15}$) but they gave more weights to the likelihood than Bayesian optimal observers would have (one tailed sign rank test compared to optimal slope $p < .00001$ except the *PwLn* combination). No statistical evidence showed that the two cognitive load groups behaved differently in the learning phase. Models which included group as a fixed effect did not fit the data better than models which did not (log likelihood ratio 10.28 compared the model with prior, likelihood and group effects to the model with prior and likelihood main effects, $p = 0.12$) and slope values between the two groups were not significantly different (median ± iqr = .74 ± .41 for serial and median ± iqr = .77 ± .25 for parallel, Wilcoxon rank sum test $p = .84$).

Validation set (**Fig 3C**) A linear mixed model that includes prior, likelihood and cognitive load group and their interactions as fixed-effect terms explained the data best (log likelihood ratio 51.05 compared to a model with prior and likelihood and their interaction as the fixed effect, $p < .001$). There was a significant three-way interaction ($p = 1.33 \text{ X } 10^{-3}$) as well as a group-likelihood ($p = 6.59 \text{ X } 10^{-07}$) and a prior-likelihood ($p = 1.81 \text{ X} 10^{-5}$) interaction. Further analyses found that the statistical result was explained by a bigger slope difference between *PnLn* and *PnLw* trials in the serial group, as compared to those of the parallel group (*PnLn* in serial median±iqr = .83±.41, *PnLw* in serial median±iqr = .20±.35; *PnLn* in parallel median ±iqr = .74 ±.26, and *PnLw* in parallel median±iqr = .61±.54). Importantly, despite the numerical difference, both groups were qualitatively Bayesian, i.e. weighting according to the reliability of prior (prior effect of linear mixed model: serial group $p = 0.02$; parallel group $p = 9.63 \text{ X } 10^{-4}$) and likelihood (likelihood effect of linear mixed model: serial group $p = 6.30 \text{ X } 10^{-19}$; parallel group $p = 5.85 \text{ X } 10^{-7}$). Like the discovery set, in both cognitive load groups, participants gave more weights to the likelihood than Bayesian optimal observers would have (one

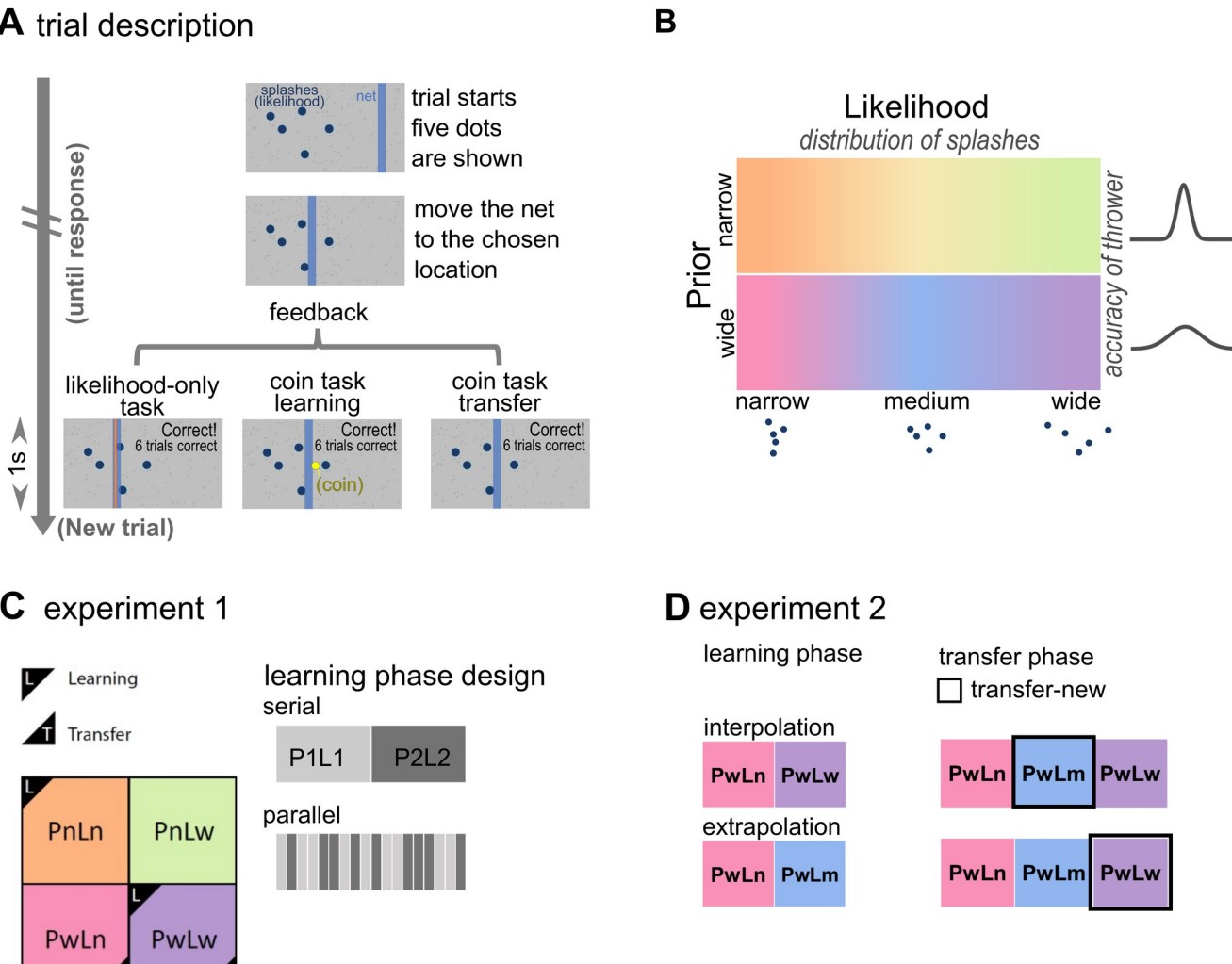

**Fig 1. Experimental design A** Trial description in likelihood-only task, coin task—learning and coin task—transfer phases. In the beginning of each trial, participants saw 5 dots which represented splashes caused by a thrower throwing a coin into the pond (grey screen). Participants moved a net (blue vertical bar) to the position where they think the centre of splashes (likelihood-only task), or the coin (coin task) is. After participants submitted a response, feedback information was displayed for 1 second on the screen. The next trial then started automatically. Feedback information differed between tasks and phases. In the likelihood-only task, the horizontal position of the real centre of splashes was displayed as an orange vertical line. In the learning phase of the coin task, the true coin position was displayed as a yellow dot every trial. An accumulated score was displayed when participants hit the coin. In the transfer phase, only an accumulated score was displayed. **B** Task design: Likelihood uncertainty was manipulated through the dot dispersion. Prior uncertainty was manipulated through the accuracy of the thrower, which participants learned from coin position feedback during the learning phase of coin task. Specific prior and likelihood combinations of the experiment 1 and 2 are explained in detail as follows. **C** Design of experiment 1. There were two priors (narrow Pn σ = .025 and wide Pw σ = .085) and two likelihoods (narrow Ln σ = .06 and wide Lw σ = .15). Participants underwent two orthogonal prior/likelihood combinations in the learning phase and then one prior coupling with both likelihoods in the transfer phase. In the figure example, the learning conditions are PnLn and PwLw (boxes with "L" ticks) while the transfer conditions are PwLn and PwLw (boxes with "T" ticks), with PwLn being the new combination of the transfer phase. For the serial group, combinations in the learning phase were delivered block-wise. For the parallel group, combinations in the learning phase were delivered in an interspersed way. Trials were always administered in an interspersed way in the transfer phase. **D** Design of experiment 2. In the learning phase, participants underwent combinations having one prior (wide Pw σ = .085) paired with two out of three (narrow Ln σ = .024, medium Lm σ = .06, wide Lw σ = .15) likelihoods. During learning, the interpolation group experienced PwLn and PwLw trials while the extrapolation group experienced PwLn and PwLm trials. All participants then undertook PwLn, PwLm and PwLw trials in the transfer phase. For the interpolation group, PwLm was the new combination. For the extrapolation group, PwLw was the new combination. All trials were administered in an interspersed way.

tailed sign rank test compared to optimal slope $p < .00001$ in all prior/likelihood combinations except the *PwLn* trials).

## Transfer phase

Discovery set We separated transfer phase data into "old" and "new" trial types. "Old" indicates those prior/likelihood combinations that individual participants had experienced during the learning phase while "new" indicates those combinations that were introduced in the transfer phase. Linear mixed models constructed to comparing the "old" combinations in the learning and transfer phase found that the model with prior and likelihood fixed effects explained slope data best. We showed that reliability-based weighting maintained in the "transfer-old" trials (linear mixed model; prior $p = 7.45$ X $10^{-3}$; likelihood $p = 1.60$ X $10^{-4}$, **Fig 3A**), meaning that behaviours remained Bayesian after the removal of location feedback. This was further supported by a non-significant phase effect (Wilcoxon signed rank test $p = .55$, evidence of phase effect $BF_{10} = .18$) when comparing the "old" combinations of the learning phase with those of the transfer phase. We then inspected the "transfer-new" trials. No group effect was observed (model including group effect v.s. model not including group effect: log likelihood ratio = 2.05, $p = .22$). There was a significant difference in slope between prior types but not between likelihood types (linear mixed model; prior $p = .01$, likelihood $p = .32$. Model including prior and likelihood effects v.s. model including only prior effect: log likelihood ratio = 1.00, $p = 0.36$) implying participants may not have transferred the knowledge in a way that ideal Bayesian observer should have (**Fig 3A**). However, it is also obvious that from **Fig 3A**, slopes varied widely between participants even in the learning phase. This noise in the data could have greatly diminished the power of detecting transfer. To remove slope variabilities caused by subject-specific prior and likelihood estimations, we computed the "transfer score" (***ts***) (**Fig 2B**, also see the **Methods** section). As intended, transfer scores did not differ between prior and likelihood conditions (three-way ANOVA; prior main effect $p = .94$, evidence of main effect $BF_{10} = .14$; likelihood main effect $p = .83$, evidence of main effect $BF_{10} = .15$; **Fig 3C** **upper panel**); supporting the "transfer score" a prior/likelihood-invariant and more valid measure of transfer. We pooled the data of different prior/likelihood combinations and found that there was significant (i.e. larger than zero; right tailed t-test; $p = 7.04$ X $10^{-12}$) but suboptimal (i.e. smaller than one; left tailed t-test $p = 1.80$ X $10^{-10}$) Bayesian transfer (**Fig 3C** **lower panel**). Transfer scores in the serial group were numerically higher than those in the parallel group but not statistically different (***ts$_{serial}$*** mean±SE = 0.59±.10, ***ts$_{parallel}$*** mean±SE = 0.46±.09; two sampled t-test $p = 0.3321$, $BF_{10} = .34$; **Fig 3C** **lower panel**).

Validation set A linear mixed model with prior, likelihood and their interactions as fixed-effect terms explained the slopes of the "old-combinations of learning and transfer phases" best (log likelihood ratio compared to a model with the prior and likelihood main effects as the fixed effect 5.20, $p = .02$). Slopes did not differ between the learning and "transfer-old" trials (log likelihood ratio between a model with phase, prior, likelihood and prior-likelihood interaction and a model with only prior, likelihood and prior-likelihood interaction 0.64, $p = 0.35$; Wilcoxon signed rank test between "old" combinations of the learning and transfer phases, $p = 0.78$). Reliability-based weighting maintained in the "transfer-old" trials (linear mixed model; prior $p = 6.88$ X $10^{-5}$; likelihood $p = 4.35$ X$10^{-7}$, **Fig 3B**), meaning that slope remained qualitatively Bayesian after the removal of location feedback. We then inspected the "transfer-new" trials. Again, no group effect was observed. There was a significant difference in sensory weighting between prior types but not between likelihood types (linear mixed model; prior $p = 2.22$ X $10^{-05}$, likelihood $p = .94$; **Fig 3B**), as has been found in the discovery set.

Transfer scores did not differ between prior and likelihood conditions (linear model with prior, likelihood, cognitive load group and their interactions as fixed-effect terms: prior main

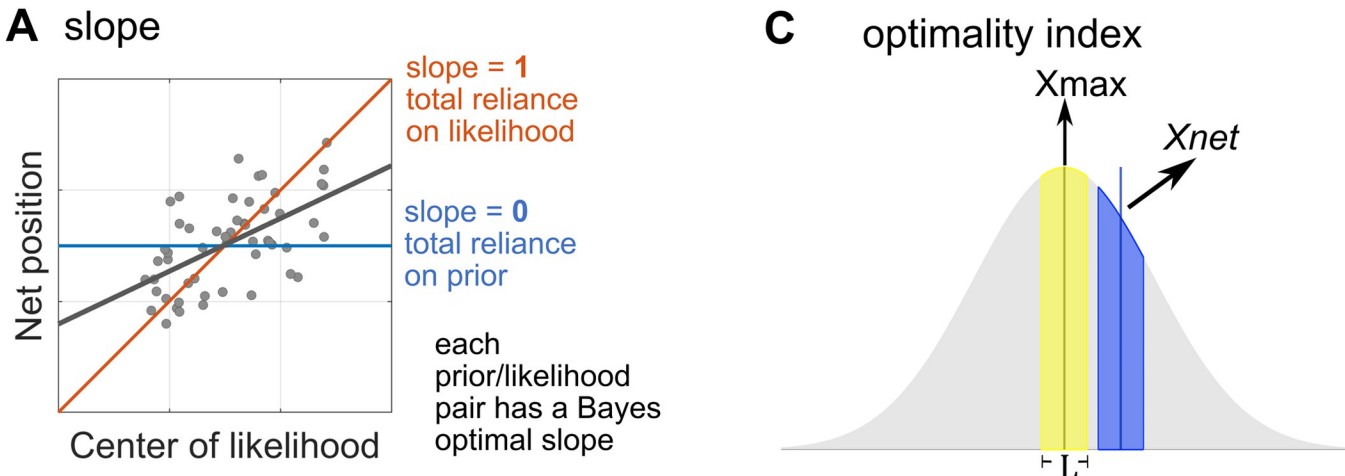

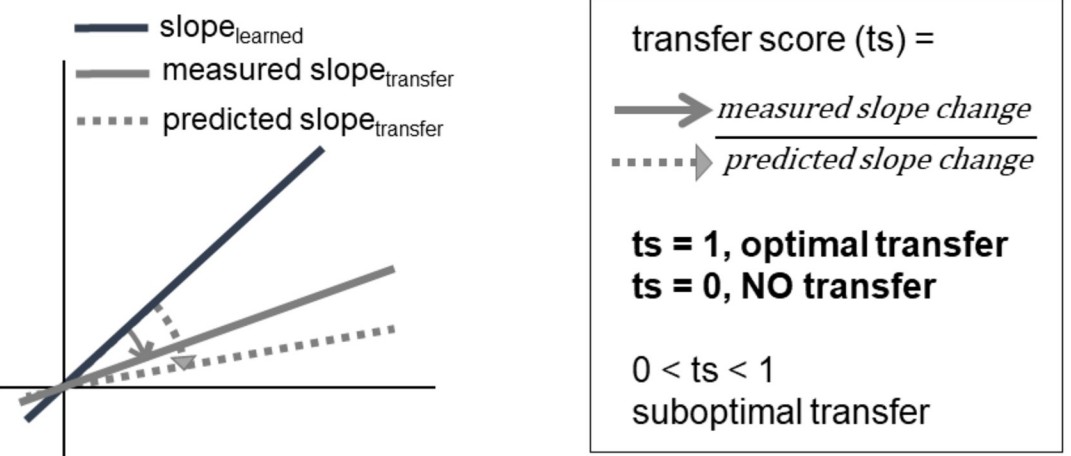

**Fig 2. visualisation of quantitative performance measures including slope (A), transfer score (B) and optimality index (C). A** Slope in the coin task was calculated by linearly regressing participants' estimated coin position over the centre of splashes. The values of slope vary between 0 and 1. A higher slope means a higher weighting on likelihood, with 1 meaning total reliance on likelihood and 0 meaning no reliance on likelihood. **B** A transfer score was calculated by normalising a measured slope change by a predicted slope change based on subject-specific prior and likelihood estimations. A transfer score of 1 means transferring following an optimal Bayesian observer model. A transfer score equals or smaller than 0 means no transfer. **C** optimality index. For each trial given the true posterior, we can compute the probability that a coin would be within the net from the chosen position ($X_{net}$), i.e., the success probability. We defined the optimality index for a trial as the success probability normalised by the maximal success probability. In the figure, this equals to the blue area divided by the yellow area. Note that here for visualisation purpose only, there is no overlap between the two areas, which may not and does not have to be the case in real measurement.

effect $p = .42$, evidence of main effect $BF_{10} = .01$; likelihood main effect $p = .39$, evidence of main effect $BF_{10} = .01$; **Fig 3D** upper panel). Nor was any interaction or group main effect (group main effect $p = .81$, evidence of main effect $BF_{10} = .01$) found. We again identified significant (i.e. larger than zero; right tailed sign rank test $p = 1.44 \times 10^{-15}$) but suboptimal (i.e. smaller than one; left tailed sign rank test $p = 7.30 \times 10^{-9}$) Bayesian transfer in pooled data. Transfer score was $.52\pm.65$ (median±iqr) in the serial group and $.50\pm.72$ (median±iqr) in the parallel group (rank sum test $p = 0.90$, **Fig 3D**).

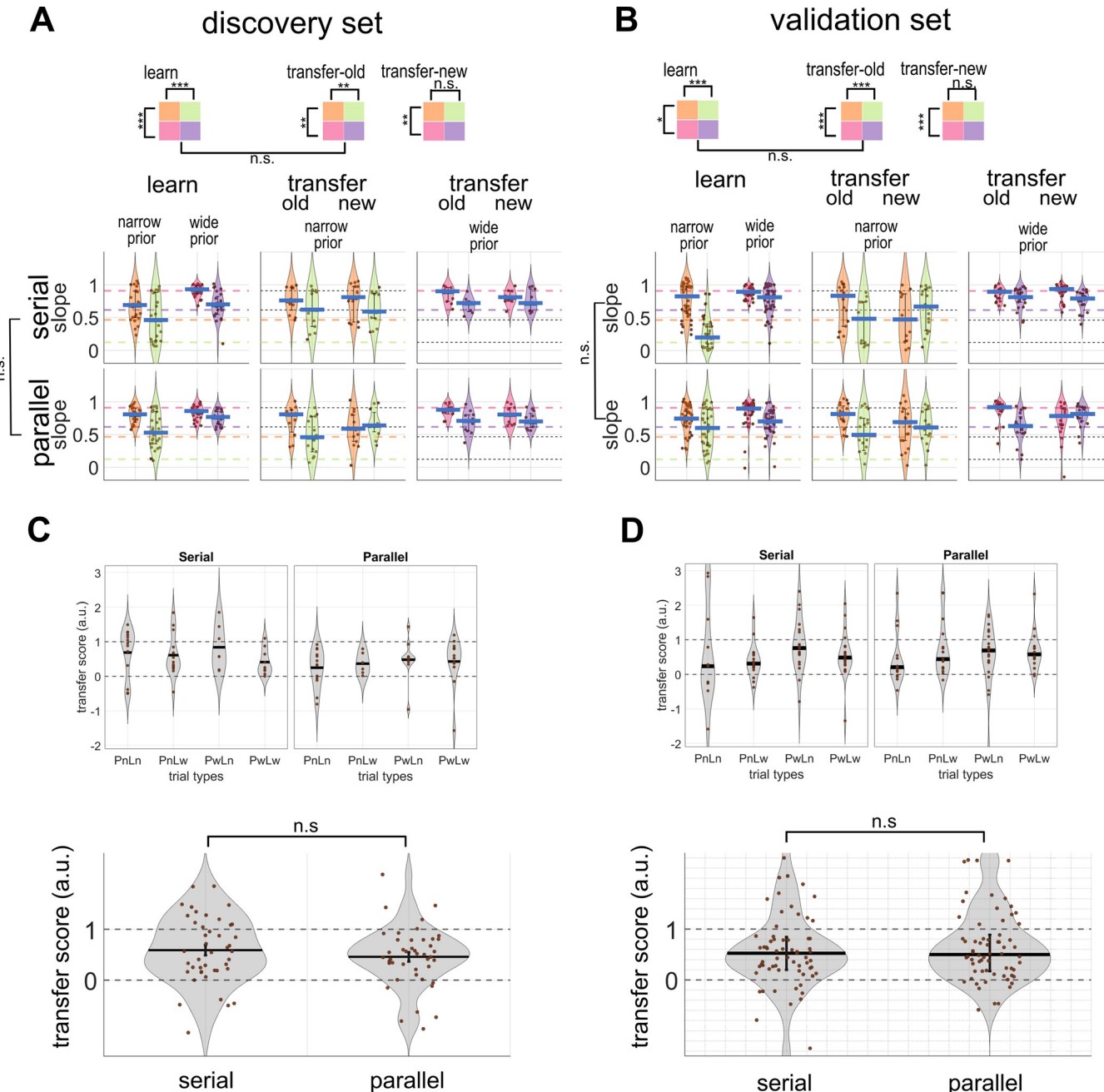

**Fig 3. Slope and transfer score in the experiment 1. slope A Discovery & B Validation set.** Violin plots show the distribution of slopes in the learning, transfer-old and transfer-new trials of experiment 1, separated by prior, likelihood and cognitive load group (upper serial, lower parallel). The central blue line shows the median. The error bar represents the interquartile range. Filled dots represent each participant. Each prior/likelihood combination is represented by orange PnLn, green PnLw, pink PwLn, and violet PwLw. Bayes-optimal values are presented as coloured dashed lines, with colours corresponding to prior/likelihood types. **transfer score B Discovery & D validation sets.** The distribution of transfer scores in the serial and parallel groups. The upper panels show distribution of transfer scores of each prior/likelihood combination, separated by cognitive load group. Lower panels merge data of different prior/likelihood combinations. The central black line and error bar in the discovery set **(Fig 3B)** represent the mean and standard error while the central black line and error bar in the validation set **(Fig 3D)** represent the median and interquartile range. *p< = .05; ** p< = .01; ***p< = .001; n.s. non-significant.

### Experiment 2

Linear mixed models were used because data were not normally distributed.

### Learning phase

<u>Discovery set</u> A full model which includes the likelihood and group effects, and their interaction explained the slope data best (log likelihood ratio as compared to a model with the likelihood and group effect but no interaction 7.89, $p$ = .02). A significant likelihood-group interaction was found ($p$ = 4.52 X $10^{-3}$). Pos-Hoc analysis showed that slopes were qualitatively Bayesian for both interpolation (median ± iqr = .98 ± .04 for *PwLn* and median ± iqr = .87 ± .18 for *PwLw*; Wilcoxon sign rank test $p$ = 1.14 X $10^{-5}$) and extrapolation (median ± iqr = .99 ± .04 for *PwLn* and median ± iqr = .98 ± .04 for *PwLm*; Wilcoxon sign rank test $p$ = .003) groups, i.e. participants decreased sensory weighting as the variance of likelihood became bigger. We concluded that the interaction was caused by a larger disparity between prior/likelihood pairs in the interpolation group than which in the extrapolation group. Like experiment 1, participants however relied on likelihoods more than Bayesian optimal observers would have in the medium and wide likelihood conditions (one tailed sign rank test compared to optimal *slope* interpolation–*PwLw* $p$ = 1.18 X $10^{-4}$, extrapolation–*PwLm* $p$ = 1.18 X $10^{-7}$, **Fig 4A**).

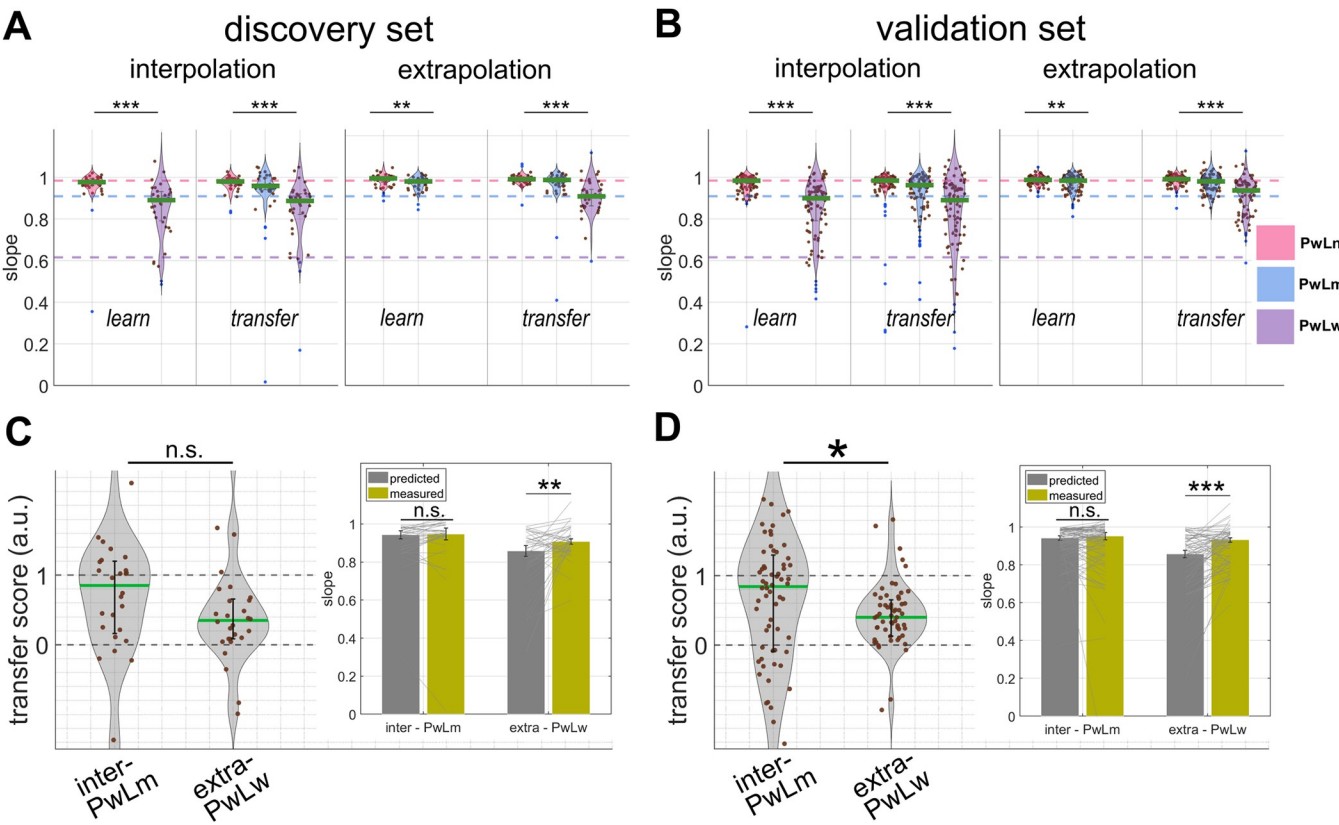

**Fig 4. Slope and transfer score of experiment 2, A discovery & B validation set** Violin-plots show the distributions of slope in the learning and transfer phases of experiment 2, separated by likelihood and group. The central green line represents the median. The vertical bar within the violin spans between the second and third quartiles. Each filled dot represents one participant. Three optimal slopes given likelihoods are presented as colour dashed lines, with pink representing PwLn, blue representing PwLm, and violet representing PwLw respectively. **C discovery & D validation set** Distributions of transfer scores in the interpolation and extrapolation groups. The central green line is the median of each group, and the vertical bar is the interquartile range. Insets showed predicted slope (gray bar) along with measured slope (olive bar) in the transfer-new trials. Note that there was no significant difference in the interpolation group. *$p$< = .05; ** $p$ < = .01; ***$p$< = .001; n.s. non-significant.

<u>Validation set</u> A full model which includes the likelihood and group effects, and their interaction explained the slope data best (log likelihood ratio 14.68, $p$ = .01). A significant likelihood-group interaction was found ($p$ = 1.09 X $10^{-4}$). Pos-Hoc analysis showed that slopes were qualitatively Bayesian for both interpolation (median ± iqr = .98 ± .05 for *PwLn* and median ± iqr = .88 ± .21 for *PwLw*; Wilcoxon sign rank test $p$ = 3.27 X $10^{-12}$) and extrapolation (median ± iqr = .99 ± .03 for *PwLn* and median ± iqr = .98 ± .06 for *PwLm*; Wilcoxon sign rank test $p$ = 2.30 X $10^{-3}$) groups, i.e. participants decreased sensory weighting as the variance of likelihood became bigger. We concluded that the interaction was caused by a larger disparity between prior/likelihood pairs in the interpolation group than which in the extrapolation group. There was an over reliance on likelihoods in the medium and wide likelihood conditions (right tailed sign rank test compared to optimal slopes; interpolation–*PwLw* $p$ = 1.30 X $10^{-13}$, extrapolation–*PwLm* $p$ = 5.34 X $10^{-13}$, **Fig 4B**).

## Transfer phase

<u>Discovery set</u> We did not find significant slope differences between transfer-old trials and learning trials (slope of "learning" trials median ± iqr = .97 ± .07 & "transfer-old" trials median ± iqr = .98 ± .07, Wilcoxon sign rank test $p$ = .38, $BF_{10}$ = .13). For all transfer trials, a likelihood-only model explained slope data best (log likelihood ratio as compared to the model with likelihood and group effects 3.96, $p$ = .06). Slopes in the transfer phase remained Bayesian (likelihood effect $p$ = 1.82 X $10^{-08}$).

Transfer scores (**Fig 4C**) were significantly bigger than zero (right tailed sign rank test $p$ = 1.29 X $10^{-7}$), implying the presence of transfer. The transfer score of the interpolation group was .85 ± 1.04 (Median ± iqr) while that of the extrapolation group was .35±.57 (Median ± iqr). The statistics is marginally significant (two side rank sum test $p$ = 0.08). We identified one difference between the two groups: For the extrapolation group, the transfer score was significantly smaller than one (one sample sign rank test, $p$ = 4.94 X $10^{-04}$, predicted versus measured slope $p$ = .005). For the interpolation group, the transfer score did not differ from one (one sample sign rank test $p$ = .12; predicted versus measured slope $p$ = 0.20). The Bayes factor ($BF_{01}$) was 0.63, showing anecdotal evidence supporting the null hypothesis.

<u>Validation set</u> Comparing "old" trials between the learning and transfer phases, a model which includes the likelihood, group effects, and their interaction explained the slope best (log likelihood ratio as compared to a model including likelihood and group but no interaction 22.65, $p$ = .01). It is noticeable that a model which also includes the phase effect failed to explain slope better than the winning model (log-likelihood ratio 4.80, $p$ = .39), supporting no changes of slopes after removing position feedback. Indeed, the difference of slope values between transfer-old trials and learning trials was negligible ("learning" trials median ± iqr = .98 ± .07 versus "transfer-old" trials median ± iqr = .98 ± .06; Wilcoxon sign rank test $p$ = .78). The post-hoc analysis for a significant likelihood-group interaction ($p$ = 1.83 X $10^{-6}$) found that the interaction was caused by a larger disparity between prior/likelihood pairs of the interpolation group (median ± iqr = .98 ± .05 for *PwLn*; median ± iqr = .88 ± .22 for *PwLw*; Wilcoxon sign rank test $p$ = 8.99 X $10^{-21}$) than which of the extrapolation group (median ± iqr = .99 ± .03 for *PwLn*; median ± iqr = .98 ± .06 for *PwLm*; Wilcoxon sign rank test $p$ = 8.61 X $10^{-6}$).

For all transfer trials, a model including the likelihood and group effects and their interaction explained the slope data better than any other models (log likelihood ratio 5.5, $p$ = .01). There was a main likelihood effect ($p$ = 3.16 X $10^{-10}$), indicating slopes in the transfer phase remained qualitatively Bayesian. Interaction was again found resulting from a larger disparity between each prior/likelihood pairs in the interpolation group than which in the extrapolation

group (interpolation *PwLn–PwLm–PwLw* (median ± iqr) .98±.04 –.95±.13 –.87±.13 versus extrapolation *PwLn–PwLm–PwLw* (median ± iqr) .99±.03 –.98±.05 –.93±.15).

Transfer scores (**Fig 4D**) were significantly higher than zero (right tailed sign rank test $p = 8.35 \times 10^{-11}$), implying the presence of transfer. Transfer score (**Fig 4D**) of the interpolation group was statistically larger than which of the extrapolation group (interpolation $0.84 \pm 1.39$ (median±iqr); extrapolation $0.40 \pm .53$ (median±iqr); two tailed rank sum test, $p = .02$; sign rank test predicted versus measured slope interpolation group $p = 0.77$; extrapolation $p = 6.07 \times 10^{-07}$) (inset of **Fig 4D**).

### Optimality index

We compared the optimality index between the first and last 10 trials of transfer-new trials (**Fig 5**). There was no difference (Fig 5 left panels; paired t-test; **Fig 5A** experiment 1-discovery $p = .81$, **Fig 5B** experiment 1- validation $p = .76$, **Fig 5C** experiment 2- discovery $p = 0.92$, **Fig 5D** experiment 2- validation $p = 0.09$), showing that participants did not use partial feedback in the transfer phase to improve performance incrementally. Similarly, no difference was

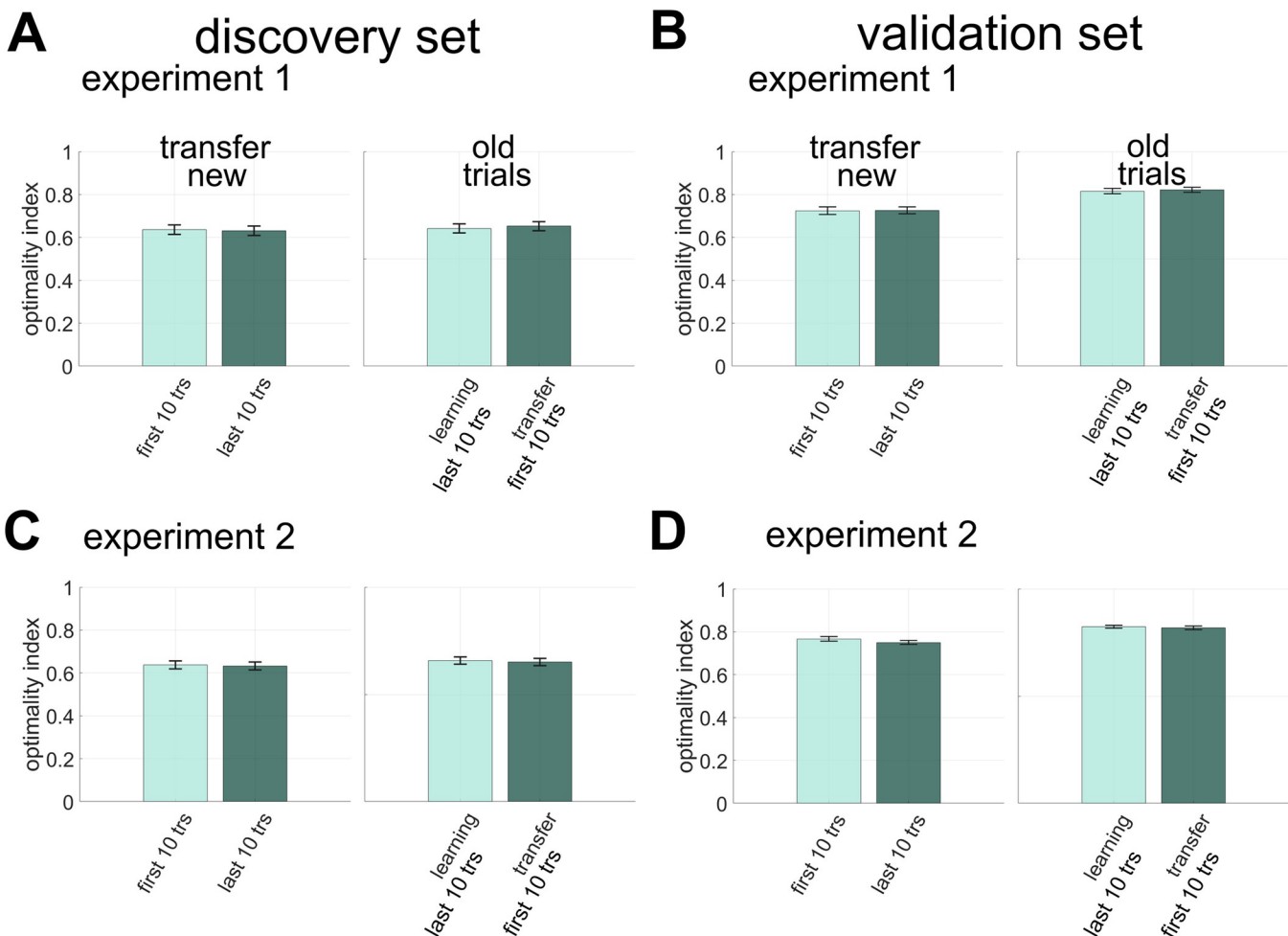

**Fig 5. Optimality index. A** discovery experiment 1 **B** validation experiment 1 **C** discovery experiment 2 **D** validation experiment 2 Bar graphs show the mean optimality index and error bars represent standard errors. In the left panel of each subplot, the mean of first 10 trials of the transfer-new (light green bar) is compared against the mean of the last 10 trials (dark green bar). In the right panel, the mean of the last 10 trials of the learning phase (light green bar) is compared against the first 10 transfer-old trials (dark green bar).

found between the first 10 transfer-old-trials and the last 10 in the learning phase (**Fig 5** right panel; paired t-test; **Fig 5A** experiment 1-discovery $p = 0.54$, **Fig 5B** experiment 1-validation $p = 0.60$ **Fig 5C** experiment 2-discovery $p = 0.83$, **Fig 5D** experiment 2-validation $p = 0.63$), further supporting that performance did not diminish after the removal of position feedback.

## Comparing with non-Bayesian models

Comparing differences between modelled and measured transfer scores (**Fig 6C and 6D upper panels**), we showed that the Bayesian model has the smallest difference to real world data in both experiments. Bayesian information criterion further showed evidence in favour of Bayesian model for both experiments (**Fig 6C and 6D lower panels**). See **S4 Fig** for simulated transfer scores of likelihood-only and linear regression models and **S5 Fig** for simulated transfer scores of exemplar model.

## Discussion

The Bayesian decision model has achieved great success in describing human behaviours across various domains. However, whether humans make explicit Bayesian computation, i.e. learning and maintaining prior and likelihood distributions to compute posterior for decision, remains an ongoing debate. Examining whether people transfer experienced priors and likelihoods to a new scenario can help answer this question. However, previous studies testing Bayesian transfer were inconclusive because they simply compared how close behaviours were to Bayes optimal in new scenarios. This is problematic because it largely depends on prior/likelihood estimation accuracy rather than transfer per se. Indeed, there are several reasons that can explain suboptimal behaviours, as previously discussed [18]. Moreover, a direct quantification of transfer was lacking. To address these limitations, we devised transfer score–a mathematical marker that quantifies transfer without being hindered by prior/likelihood estimation biases. By applying the transfer score to the coin task [35], we found significant albeit suboptimal transfer in two perceptual decision studies. For each study, our results were replicated across two independent datasets (discovery and validation sets), suggesting robust replicability.

   In the first study, we manipulated the way in which two location priors were learned and observed their effects on Bayesian transfer. The purpose was to better understand how much cognitive load can restrict the use of Bayesian computation. One argument against explicit Bayesian computation in human cognition is its complex calculations, and hence the greater demand it imposes on the brain compared to non-probabilistic computations [15,37]. A previous study [27] tested Bayesian transfer when people concurrently learned two position priors. They found that weighting change in accordance with the newly introduced likelihood uncertainty did happen, but not sufficiently, and it only happened until an explicit instruction informing two different prior uncertainties was given. Thus, it was considered "suboptimal transfer". We hypothesised if simultaneous multiple prior learning was the cause of such weak transfer they observed, sequentially presenting priors may rescue transfer performance as suggested by another study [25]. We compared sequential (similar to [25]) and concurrent (similar to [27]) prior learning side by side. Our transfer scores showed that the two conditions led to equivalently suboptimal transfer. Thus, we did not find evidence supporting that simultaneous learning of two spatial priors prompted the use of non-Bayesian strategies to impede transfer. Interestingly, even though we explicitly instructed participants that our two priors had different levels of uncertainty, we could not identify transfer in the slope measurement as defined and observed in Kiryakova and colleagues [22]. We concluded that these suboptimal

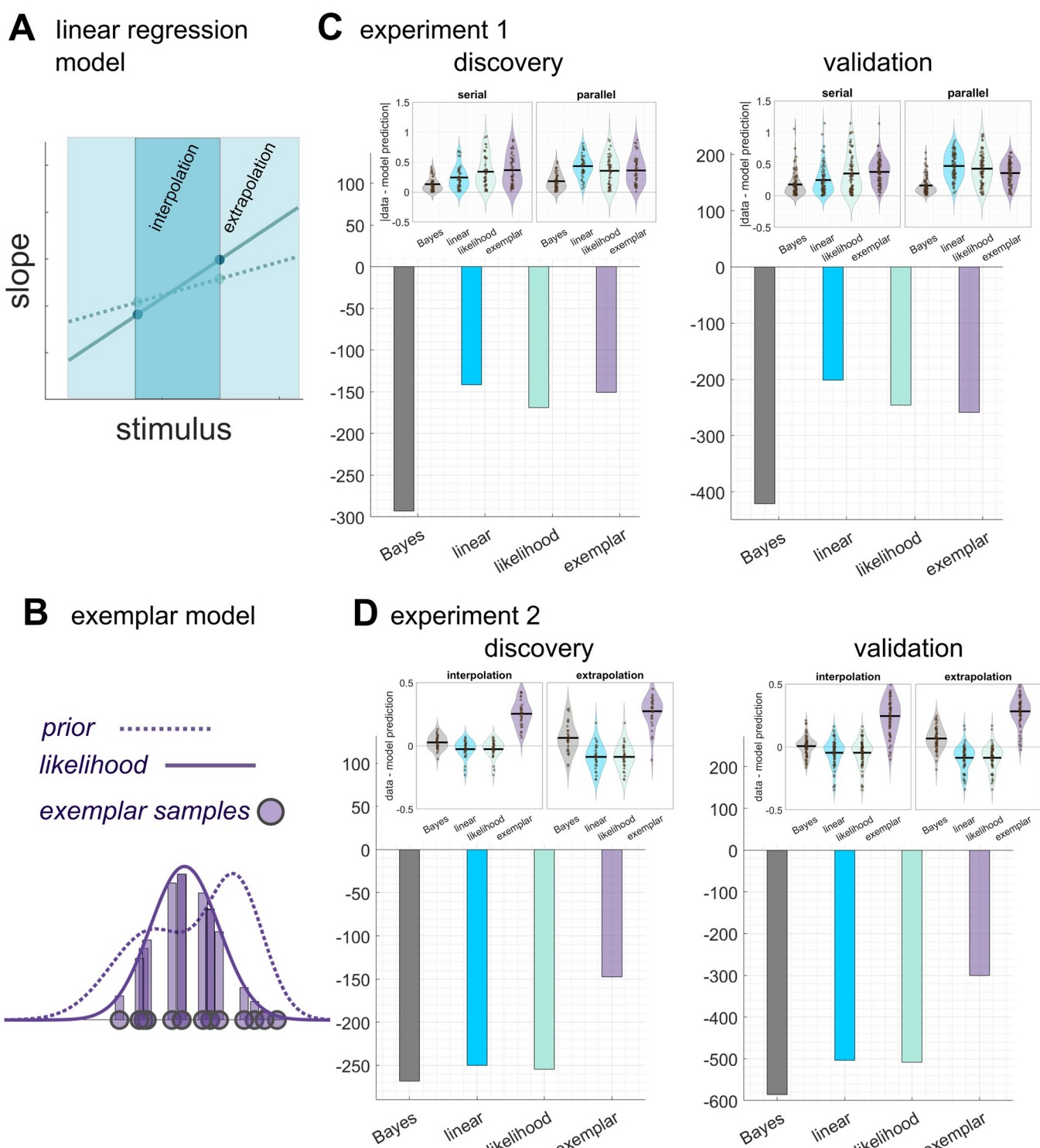

**Fig 6. Computational models and comparison of model fitness with Bayesian decision model A** schematic illustration of a linear regression model for the coin task. An optimal linear mapping between scatters of splashes and slopes (solid line) will produce optimal extrapolation even if no true generative process is represented. In reality, because of noisy slope estimations (dotted line), interpolation (darker green zone) would be closer to optimal than extrapolation (light green zone). **B** schematic illustration of an exemplar model for the coin task. Instead of representing the parameters of prior distribution, participants retrieved exemplar samples (violet circles) acquired from the learning phase. Samples are then weighted (bars representing weighting) by the gaussian likelihood of splashes to infer the coin position of a given trial. **C (experiment 1) & D (experiment 2)** Model comparison using Bayesian information criteria (BIC). Colour code: grey Bayesian model, blue linear regression model, green likelihood-only regression model, and violet exemplar model. The smaller BIC values indicate the better models. Insets showed differences between modelled and measured transfer scores.

behaviours found in past studies and ours are more likely a manifestation of imprecise uncertainty estimation rather than the execution of non-probabilistic computation.

The critical question is how much people can generalize knowledge to unseen data in any given task. We addressed the question head on in the second study. We compared the transfer score between generalization within (interpolation) and outside (extrapolation) the range of experienced sensory information (likelihoods) when performing our visuospatial task. Again, there was significant but suboptimal transfer in both conditions. Between the two conditions, people had a lower transfer score in the extrapolation and this difference was statistically significant in the (larger) validation dataset. These results indicated that generalization is better when new data points are within the context of past observations. This is consistent with what Kiryakova and colleagues [27] have shown. However, our design shielded any observed effect from the influence of imperfect uncertainty estimation (in the approximation algorithm) and is hence more robust. Moreover, unlike previous studies, we quantified the degree of transfer.

Taking findings of the two experiments, we ask, what is the most possible computation strategy for the coin task? There was no evidence supporting the use of a heuristic strategy in experiment 1. A stable optimality index also suggests participants did not make decisions using model-free (error-based) learning. A Bayesian observer that acquires a true generative model of the task, however, would have behaved similarly between interpolation and extrapolation, which was not what we observed in the second experiment. The results of the two experiments seemed to be at odds, with the first supporting and the second casting a doubt on Bayesian computation, albeit with knowledge transfer. Due to the Gaussian noise feature of the coin task, the optimal policy is a linear mapping between stimuli and response (Eq 1). We need to entertain the possibility that participants could have capitalised on this feature and learned a mapping for one stimulus type without forming a generative model. One might think that a linear-mapping learner should paradoxically generalize equally well in interpolation and extrapolation. We argue that due to the biased estimates observed in our data, it is only reasonable to see lower transfer in extrapolation if our participants used a linear mapping policy (**Fig 6A**). We constructed such linear models. While we did observe better transfer in interpolation than in extrapolation, Bayesian Information Criterion did not support this hypothesis (**Fig 6C and 6D**). Along with a total absence of transfer in extrapolation (**S4C and S4D Fig**), it is unlikely that people utilise a linear mapping strategy for the coin task. We also built an exemplar model to understand if people could also have used exemplar memory acquired from the learning phase for transfer and failed to find evidence to support so. Together, it is possible that participants acquired a probabilistic recognition model linking contextualised sensory inputs to corresponding policies [21,22]. The model approximates posterior where it has been covered by training data with a surrogate distribution. It would generalize well enough within, but not as well as outside past experience.

One limitation of transfer score is that its accuracy is dependent on the accuracy of $\sigma_{Pi}^2$ (Eq 4). When slopes are close to 1, there could be arbitrarily large prior variances (Eq 4) leading to unrealistic transfer scores. In this study we have made great efforts to reduce this negative effect (as described in **S1 Text** 'Handling arbitrarily large slopes' and 'Outlier participant exclusion criteria'). Importantly, by scrutinising the data (see **S1 Text** and **S2 Fig**), we showed that our transfer score patterns are unlikely to be subject to this limitation. However, we acknowledge that attrition of participants during the process of transfer score outlier exclusion is a shortcoming. This can potentially be improved in the future by choosing prior/likelihood combinations which are unlikely to produce large slope values.

Our studies showed that people may not perform decisions under uncertainty by computing the full generative model. This is in line with many neurocomputational theories

formulating that approximation algorithms are needed for human Bayesian cognition [22,38–40]. We showed that perceptual decisions use Bayesian approximation with knowledge transfer, albeit suboptimal. We also demonstrated that we can more effectively validate transfer using a rigorous mathematical definition of transfer score, that can be applied to various probabilistic inference tasks and advance our computational understanding of human behaviours. It would be especially valuable to test transfer in a similar perceptual decision task that uses likelihoods presenting realistic noise (such as luminance). Based on the abovementioned bounded-optimal hypothesis, transfer will be close to optimal in both interpolation and extrapolation case given that our brain has been trained by such types of likelihood through life-long experience (i.e., hardwired priors) while linear mapping learners are likely to behave similarly to the current study.

More studies are needed to confirm the exact computation underpinning decisions. Recently, another paradigm [41] was proposed for scrutinising whether humans make inference using explicit Bayesian computation. The paradigm uses multiple prior-likelihood pairs in the Bayesian decision model which lead to an identical decision policy and investigates the learning dynamics when new component (i.e. prior-likelihood) pairs being introduced to discriminate explicit Bayesian learners from policy learners. In short, right after a transition to a new pair, explicit Bayesian learners' performance would temporarily deteriorate as they relearn the new prior/likelihood while policy learners would maintain the same level of performance. We previously argued that some tasks are more sensitive to transfer manipulations while others are more sensitive to component-pair manipulations and the two can be complementary in validating true Bayesian learners behaviourally [18]. Future work could make use of these two complementing paradigms to systematically inquire about the exact process model that supports different types of decision under uncertainty.

## Methods

### Participants

The online research was approved by the University of Melbourne research ethics committee (research ethics project reference number 20592). Participants were recruited using the University of Melbourne psychology research participation pool and Prolific online survey platform (prolific.co). All participants completed self-reported questionnaire to confirm that they had normal or corrected-to-normal visual acuity, and no history of neurological, psychiatric disorders or substance use, and gave written informed consent prior to taking part in the study. Participants were compensated with course credits or £5 per hour for their time.

For experiment 1, 102 adults (74 females, age mean ± SD = 19.81 ± 3.92 yrs) were recruited for the discovery study. Among these participants, data from 6 participants were excluded entirely, 1 for not meeting the inclusion criteria and 5 for poor data quality (exclusion criterion based on data quality see below **Data analysis**), resulting in a final sample of 96 participants (48 for each group). For the validation study, 170 participants (120 females, age mean ± SD = 19.90 ± 5.60 yrs) were recruited. Among them, 11 participants were excluded for poor data quality, resulting in a final sample of 159 participants (80 for serial group and 79 for parallel group). There was neither a significant age nor gender difference between the discovery and validation set.

For experiment 2, 99 participants (66 females, age mean ± SD = 20.36 ± 3.91 yrs) were recruited for the discovery study. Among these participants, data from 13 participants were excluded for history of neurological or psychiatric disorders and 11 for poor data quality, resulting in a final sample of 75 participants (35 for interpolation and 40 for extrapolation group). For the validation study, 174 participants (136 females, age mean ± SD = 19.71 ± 3.28

yrs) were recruited. Among them 10 participants were excluded for poor data quality, resulting in a final sample of 164 participants (82 for each group). There was neither a significant age nor a gender difference between the discovery and validation set.

Our power analysis was based on the findings of the discovery data (R software: "SSDbain" and [42], indicating that, to increase the power to 80% to confirm a moderate effects at $a = 0.05$ for a one-sided t test, 80 participants would be required for each group.

## Experiments

All experiments were designed using PsychoPy (PsychoPy 2020) and launched online through the Pavlovia platform. Stimuli were displayed on participants' own screens. The minimum requirement for a screen resolution was 1172x553 pixel.

## Coin task

All stimuli were in an arbitrary unit that defines the horizontal location of the left and right edges of the screen as -0.5 and 0.5. Participants were instructed to view the screen as the surface of a pond and to locate an unseen coin that a person had thrown into the water (**Fig 1A**). At the beginning of each trial, participants saw five dots (diameter = 0.01) which represent the "splashes" caused by a coin dropping to the pond. The horizontal positions of these five dots were drawn from a Gaussian "likelihood" distribution with a mean of the horizontal coin position in that trial and a standard deviation assigned from one of the three values, ($\sigma_L = 0.0024$, 0.06 and 0.15). In each trial, the horizontal location of the coin was drawn from a Gaussian "prior" distribution which centres at the middle of the screen (mean = 0) and has a standard deviation $\sigma_p$ of either 0.025 or 0.085 (**Fig 1B**). Participants' task was to catch the coin by moving a vertical blue bar (the "net", width $l = 0.02$) to their estimated hidden coin position and then click on the "space" button to indicate the answer. As the height of the bar equalled the height of the screen, vertical locations of splashes and coin made no difference to participants. There was no temporal deadline, so participants had all the time they needed to submit their response. After responding, participants received feedback information for one second before the next trial started automatically. There were two phases: learning and transfer phases, in the coin tasks (see more information about the two phases in **Methods** - **Experiment specific details**). Feedback information of the two phases differed as follows (**Fig 1A lower panel**). In the learning phase, participants received trial-by-trial coin location feedback, which was displayed as one yellow dot (diameter $d = 0.01$) along with splashes. If there was an overlap between the net and the coin, this trial was deemed a correct trial and participants' scores increased by one point. Coin location feedback was given in every trial and for every successful trial the accumulative score was displayed along with it. In the transfer phase, only scores but not coin positions were given. There were minor differences in how the scores were shown between the discovery and validation study during the transfer phase. In the discovery set, scores were shown in all corrective trials. In the validation set, participants received a summary of their total correct trials every 15 trials during the transfer phase. In both sets, at the beginning of the transfer phase, we informed participants about this change of feedback. However, we also reminded participants that throughout the experiment the goal was to be as accurate as possible in locating the coin, irrespective of feedback.

## Likelihood-only task

We also administered a likelihood-only task to measure how good participants were at locating the centroids of dot clouds at each level of likelihood uncertainty ($\sigma_L = 0.0024$, 0.06 and 0.15) in the absence of prior information. In each trial, participants saw five "splashes" on the screen.

They were instructed to move the "net" to where they thought the "centre" of the 5 dots on the horizontal axis (x-axis) was. After responding, the true horizontal location of the "centre" was revealed by an orange vertical bar (Fig 1A left **lower panel**) which had a height equalling the height of the screen and a width of.006. Previously [43] it was found that estimations of splash centres were biased by priors when people undertook the likelihood-only task after the coin task. Therefore, we administered the likelihood-only task before the coin task in the experiments.

## Visual memory task

In both experiments, a Corsi block-tapping test [44] was used to assess participants' working memory before they started the likelihood-only task. The task began by flashing several blocks on the screen. Participants were required to respond by tapping the block either in the forward or backward order of flashing. The test started with three flashes, the number of flashes increased by one after a participant gave one correct answer and decreased by one after a participant gave two successive incorrect answers. There were 16 working-memory trials in total, eight in a block of serial order followed by eight trials in a block of reverse-serial order. We used this task as a screening task. Participants who failed to correctly respond to 3 flashes were excluded from further analysis.

## Experiment 1 specific details (Fig 1C)

Participants were randomly assigned to a serial or parallel learning group. The two groups only differed in the presentation order of trial types (prior-likelihood pairs) in the learning phase. Both groups completed the likelihood-only task, followed by the learning, and then transfer phase of the coin task. The likelihood-only task constituted of total 80 trials, interspersing 40 of each likelihood distribution (narrow $\sigma_{Ln}$ = 0.06 and wide $\sigma_{Lw}$ = 0.15). The learning phase of the coin task constituted of 400 trials, 200 trials of two sets of prior (narrow $\sigma_{Pn}$ = 0.025 and wide $\sigma_{Pw}$ = 0.085) and likelihood (narrow $\sigma_{Ln}$ = 0.06 or wide $\sigma_{Lw}$ = 0.15) combinations. Overall, there were four possible prior-likelihood combinations, i.e. *PnLn*, *PnLw*, *PwLn* and *PwLw*. For each participant, the combinations were always orthogonalized such that one likelihood only paired with one prior but not the other during learning. Prior-likelihood combinations were counterbalanced across participants. For the serial learning group, participants learned one combination in trial 1–200 (4 blocks of 50 trials) and then the other in trial 201–400. For the parallel learning group, the two prior-likelihood pairs were interleaved throughout the leaning phase. We used a graph instruction to inform participants that there are two throwers with different levels of accuracy hitting a coin at every block beginning. During a trial, throwers were represented by two splash colours–green and blue, with the colour and prior pairing counterbalanced across participants. In the transfer phase, both groups experienced 180 trials interspersing 90 of each likelihood distribution paring with one prior.

## Experiment 2 specific details

Participants completed the likelihood-only task, followed by the learning, and then transfer phases of the coin task. In the likelihood-only task, there were 90 trials with 30 trials of each likelihood distribution (narrow $\sigma_{Ln}$ = 0.024, medium $\sigma_{Lm}$ = 0.06, and wide $\sigma_{Lw}$ = 0.15) interspersing over three 30-trial blocks. Only one prior ($\sigma_{Pw}$ = 0.085) was used for experiment 2.

Participants were randomly assigned to an "interpolation" or "extrapolation" group for the coin task (Fig 1D). For the interpolation group, participants were presented with 100 *PwLn* and 100 *PwLw* trials in the learning phase. For the extrapolation group, participants experienced 100 *PwLn* and 100 *PwLm* trials in the learning phase (Fig 1D). During the transfer

phase, all participants encountered *PwLn*, *PwLm* and *PwLw* trials pseudorandomly (225 trials, 75 trials for each prior/likelihood pair). Therefore, for the interpolation group the new combination in the transfer task was *PwLm* while for the extrapolation group was *PwLw* trials.

## Data analysis

All the data analyses were performed using R studio (R 4.1.1) and Matlab (Mathworks release 2021b). We used a median absolute deviation (MAD) to identify and exclude outliers in trials and individual participants [45]. It would exclude any data points that were 3 MAD away from the median of a participant's data. Details of specific outlier exclusion criteria for slopes, internal estimations of priors, predicted slopes and transfer scores and participant numbers for all the analyses are detailed in **S1 Text** and **S2 Table**.

## Estimating likelihood reliance (slope) using Bayesian theory

According to the Bayes rule, an optimal estimation of the coin location ($X_{est}$) is:

$$X_{est} = \frac{\sigma_L^2}{\sigma_P^2 + \sigma_L^2}\mu_P + \frac{\sigma_P^2}{\sigma_P^2 + \sigma_L^2}\mu_L \tag{1}$$

That is, an optimal estimated position is a weighted average of prior mean $\mu_P$ and likelihood mean $\mu_L$ and the weighting is based on the relative precision (= the inverse of variance $\sigma^2$), with $\sigma_P^2$ being the prior variance and $\sigma_L{}^2$ being the variance associated with the likelihood, which can be estimated by $\sigma_L{}^2$ = likelihood variance/number of dots (5 in our case). We can compare a participant's likelihood weight against this optimal decision to learn if behaviours are optimal. Similar to [35], for each prior/likelihood combination in each participant, we used the polyfit.m function in Matlab to fit chosen net locations $x_{net}$ a linear function of cue centre positions $Xnet = intercept + \beta \cdot \mu_L$. We discarded the first 50 trials (experiment 1)/25 trials (experiment 2) of every prior/likelihood pair (based on ref [35] and the findings of **S1 Fig**) to minimize the effect of the initial learning phase. The slope of the regression line (= $\beta$) indicates how much participants relied on likelihood, i.e. sensory weight (**Fig 2A**), and is expected to equal ($\frac{\sigma_P^2}{\sigma_P^2 + \sigma_L^2}$) for a Bayesian optimal observer. The closer the slope is to one, the more a participant relies on the likelihood. The closer the slope is to zero, the more a participant relies on the prior.

In our data, fitted linear functions had minuscule non-zero intercepts (mean±std 0.0027 ±0.0078, see **S1 Table** for group means of each prior/likelihood combinations). Similar findings have been found [25] and these previous works showed small biases had no effects on Bayesian slopes. To ensure our results were unaffected by the bias, we computed transfer scores using slopes acquired when forcing the regression line intercept to zero (supplementary **S3 Fig**). We confirmed that transfer behaviours in both experiments were not affected by non-zero intercepts.

## Predicted slope in the transfer phase based on proxies for subject-specific prior and likelihood variances

Numerous studies using the coin task and similar paradigms [25,35,43,46–48] consistently demonstrated that people integrate prior and likelihood information in a qualitatively Bayesian fashion. However, empirical slope values are often different from optimal Bayesian values. This difference increases the measurement noise of slope values in the transfer phase. Therefore, a slope value which deviates from Bayes optimal during transfer may either be a manifestation of suboptimal Bayesian behaviours [18], or a result of a non-Bayesian strategy. Simply measuring slopes cannot discriminate the two, so it is an ineffective way of determining

transfer. Instead, we postulate a measure inferring knowledge transfer (as defined in Eq 5 in the next section: transfer score) that relies on proxies for subject-specific prior and likelihood variances as described below.

Studies have shown that instead of estimating coin locations based on experimenter-imposed prior and likelihood variability, people's judgments of the coin location are more likely weighted by the participants' estimated variability of likelihood and prior (i.e. perceived errors to them) [27,43]. To express this concept mathematically, we changed (Eq 1) to

$$\dot{X}_{est} = \frac{\sigma^2_{Li}}{\sigma^2_{Pi} + \sigma^2_{Li}} \mu_P + \frac{\sigma^2_{Pi}}{\sigma^2_{Pi} + \sigma^2_{Li}} \mu_L \tag{2}$$

, which shows that a participant's estimated position $\dot{X}_{est}$ is again a weighted average of prior mean $\mu_P$ and likelihood mean $\mu_L$. However, in contrast to Eq 1, weights are based on estimated subject-specific behavioural precision for the prior and likelihood. Specifically, $\sigma^2_{Pi}$ evaluates how precisely participants can locate a hidden coin in the absence of splashes and $\sigma^2_{Li}$, a proxy for subject-specific likelihood, evaluates the participants' precision of estimated centres of likelihood. Importantly, we can approximate $\sigma^2_{Pi}$ and $\sigma^2_{Li}$ from individuals' likelihood-only task and learning phase data to predict transfer-phase slopes and then compare against real data in the transfer phase. In the following, we explain how we did so step-by-step.

Firstly, we calculated $\sigma^2_{Li}$, the variance of estimates of the mean ($\mu_{Lest}$) relative to the true mean of the splashes ($\mu_L$) in the likelihood-only task [43,49,50]

$$\sigma^2_{Li} = \frac{\sum \left( \mu_{Lest} - \mu_L \right)^2}{n_{Trials}} \tag{3}$$

The number of trials ($n_{Trials}$) for each likelihood condition was 40 in the experiment 1 and 30 in the experiment 2. Note that $\sigma^2_{Li}$ not only represents sensory precision but also captures a combination of errors of perception (dot-location-measurement), computation (centroid-computation), and action (response/motor). Importantly, this proxy for subjective likelihood variance has been shown to explain slope data better than using experimenter-imposed (by design) likelihood variance [27]. Secondly, $\sigma^2_{Pi}$ is computed as a combination of $\sigma^2_{Li}$ and sensory weight, which is derived by rearranging the equation ($slope = \frac{\sigma^2_{Pi}}{\sigma^2_{Pi} + \sigma^2_{Li}}$) as follows [43,49,50]

$$\sigma^2_{Pi} = \frac{\sigma^2_{Li} * slope}{(1 - slope)} \tag{4}$$

Thirdly, assuming $\sigma^2_{Pi}$ and $\sigma^2_{Li}$ were stable between learning and transfer phases, for each participant the $\sigma^2_{Pi}$ and $\sigma^2_{Li}$ of transfer-new trials were then plugged back into the equation ($slope = \frac{\sigma^2_{Pi}}{\sigma^2_{Pi} + \sigma^2_{Li}}$) to acquire a predicted transfer-phase slope.

## Transfer score (Fig 2B)

Expanding on the predicted transfer-phase slope, we developed a measure, called the transfer score, to quantify how well people generalize knowledge about prior uncertainty that they acquired in the learning phase into the transfer phase. A transfer score (*ts*) is defined as follows:

$$ts = \frac{measured\ slope\ change}{predicted\ slope\ change} = \frac{(measured\ slope_{transfer} - slope_{learned})}{(predicted\ slope_{transfer} - slope_{learned})} \tag{5}$$

The transfer score compares the measured "change" of slopes against a predicted change of slopes. In experiment 2, for each participant we would have acquired two measured slope changes using the two prior/likelihood combinations during the learning phase. We took their mean to calculate the transfer score of each participant. Note that transfer performance peaks at the value of one, (i.e. "Bayesian optimal" transfer score = 1). A score equal to or smaller than zero means no transfer, and a score between zero and one indicates suboptimal transfer. Crucially a transfer score _systematically_ larger than one is, from the Bayesian point of view, a deviation from optimum, not a "supra-optimal" transfer (e.g., excessive slope adjustments, examples can be seen in supplementary **S4 Fig**). In our study there were also cases where even Bayesian computational strategies could show larger than 1 transfer score because of arbitrarily big subject-specific prior estimates (supplementary **S2 Fig**). These undesirable artefacts were caused by noise introduced by the fitting procedure or measurement noise. We specifically devoted sections in the supplementary information (**S1 Text** supplementary methods: 'handling arbitrarily large slopes' & 'outlier participant exclusion criteria') to explain how we minimised their effects on our data.

## Optimality index (Fig 2C)

We wanted to understand whether people's weighting of likelihood changed in the absence of feedback on the coin position during the transfer phase. For transfer-old trials, we wanted to confirm that the performance level did not drop right after the disappearance of feedback. For transfer-new trials, we wanted to show that there were no significant changes (improvement or decline) of performance between the beginning and the end of the transfer phase. In this context, the multiple prior-likelihood combinations within and between participants became confounding factors, we therefore adopted a general measure of performance: the "optimality index", that is applicable beyond the same prior-likelihood combinations from [26]. The rationale of optimality index is as follows. For every horizontal position on the screen **x**, we can calculate the probability of hitting the true coin location $p_{hit}$ given a participant placing the net on **x**. The analytical solution of $p_{hit}$ based on the mean **μ** _and standard deviation of_ **σ** of the true Gaussian posterior for each prior and likelihood combination is

$$p_{hit} = \frac{1}{\sigma * \sqrt{2\pi}} \int_{x-\left(\frac{L}{2}\right)}^{x+\left(\frac{L}{2}\right)} e^{-\frac{(x-\mu)^2}{2\sigma^2}} dx \qquad (6)$$

_L_ equals the width of the net _l_ plus the diameter of the coin _**d**_. $p_{hit}$ is visualised in the **Fig 2C** as the area under the probability density function curve. The hitting probability is maximal (= $max(p_{hit})$, illustrated as the yellow area in the **Fig 2C**) when **x** equals **μ**. The optimality index allows us to track performance on a trial-by-trial basis and it is defined as $p_{hit}(x_{net})/max(p_{hit})$. For our purpose, we compared the average of the last 10 learning phase trials and the first 10 transfer-old-trials as well as the first and last 10 of transfer-new trials.

## Models

We examined how well non-Bayesian strategy models fit transfer behaviours as compared to an ideal Bayesian model. Here we described the non-Bayesian models. Model comparison was conducted using the Bayesian Information Criterion (BIC).

**Linear regression models.** Rather than computing slopes using the Bayesian decision model, people can utilise a strategy of linearly mapping between estimated slopes and prior and likelihood reliabilit*y (= inverse of variance)*. The linear mapping can be expressed as the following equation: $slope = \beta_0 + \beta_1 \cdot \frac{1}{\sigma_{P_i}^2} + \beta_2 \cdot \frac{1}{\sigma_{L_i}^2}$. For each participant, we first evaluated the

coefficients $\{\beta_0, \beta_1, \beta_2\}$ using the learning phase data. We then plugged prior and likelihood variances in the transfer phase into the parametrised equation to predict transfer-phase slopes. In a special case of $\beta_1 = 0$, the model only accounts for a linear mapping between slope and likelihood reliability and we called it a likelihood-only model. Transfer scores predicted by these linear models are shown in the **S1 Text** supplementary results and **S4 Fig**. BIC was used to compare the fitness between Bayesian and linear models.

**Exemplar model.**   We created an exemplar model of the coin task based on [51,52]. Let us assume participants acquire the total past observed trials of coin locations during the learning phase as an exemplar set $X^*$. This means that priors are represented as exemplar memory instead of a probability distributions $p(x)$. Upon seeing the splashes in a trial $i$., $N$ samples of exemplar $x^*$are drawn from $X^*$. Each drawn sample is then weighted by the splash distribution (i.e. likelihood distribution $p(\mu_{Li}, \sigma_L^2)$ where $\mu_{Li}$ is the centre of the splashes and $\sigma_L^2$ is the likelihood variance in trial $i$). We then take the normalised weighted sum of the values of sampled exemplars to get the posterior estimate $\dot{x}_i$ of the true coin location $x_i$. This exemplar model can be expressed as the equation below. Here $f(x_j^*, X^*)$ is simply a function describing how samples are drawn from the exemplar set.

$$\dot{x}_i = \frac{\sum_{j=1}^{n} f(x_j^*, X^*) p(x_j^* | \mu_i, \sigma_L)}{\sum_{j=1}^{n} p(x_j^* | \mu_L, \sigma_L)} \tag{7}$$

In the coin task, we assumed samples were randomly drawn from all coin positions displayed in the learning phase. We implemented the model when the number of sampled exemplars size is small ($N = 5$) and moderate ($N = 20$). Modelled data of slope and transfer score are shown in the **S1 Text** supplementary results and **S5 Fig**.

## Statistical analysis

We used the Kolmogorov-Smirnov test to evaluate the normality of data. Parametric tests including mixed-design ANOVA and t-test were used for normally distributed data. For data which violated the assumption of normality we used fitlme.m in Matlab to perform a linear mixed effects analysis and Wilcoxon tests to compare the medians. The significance level of all tests was 0.05. Where appropriate, Bayes factors were also reported in support of evidence for the null hypothesis.

## Supporting information

**S1 Text. Supplementary Methods.** Analysing sensory weights learning across time in the learning phase. Handling arbitrarily large slopes: removal of larger than 1 slope values or logistic transformed slope. Outlier participant exclusion criteria. **Supplementary results.** Untransformed versus logistic transformed subject-specific prior values. Linear regression and likelihood-only modelled transfer scores. Simulated responses using the exemplar model. **Supplementary references.**
(DOCX)

**S1 Fig. Time course of slopes in the learning phase.** The instantaneous slope of each trial was calculated by regressing the following ten trials (including the current trial), except in trial 192–195 for experiment 1 and trial 92–95 for experiment 2 slope was computed by regressing the following six trials. The last 5 trials of each condition were not shown as to avoid biased slope presentations due to scarce data points. Coloured lines are group means and shades represent standard errors. The vertical line in each plot separates trials that were excluded (before)/included (after) for learning phase slope calculation in the main analysis. A discovery

experiment 1 B validation experiment 1 C validation experiment 2 D validation experiment 2. (TIF)

**S2 Fig. Subject-specific prior uncertainty (variance and standard deviation)** A Scatter plots of prior estimates, experiment 1 discovery set. Left panel: serial learning group; right panel: parallel learning group. The X-axis represents untransformed prior and the Y-axis represents logistic-transformed prior. The order of estimates among the same experimenter designed prior categories are largely maintained after transformation but the distributions of values are compressed, as expected. B Bar figures to compare the group means of un-transformed (left panel) and transformed (right panel) subject-specific priors of experiment 1, discovery set. Error bars are standard errors. Dash lines are the experimenter designed standard deviations of narrow (= 0.025) and wide (= 0.085) priors. Figures show that the order of different priors only maintained in the untransformed estimations. They also show that the numerical values are generally smaller than the experimenter designed values. C Bar figures to compare the group means of un-transformed subject-specific priors of the experiment 2. Dash lines are the experimenter designed standard deviations of the prior (= 0.025). Overestimations were observed, especially in the extrapolation group. Left panel: discovery set; right panel: validation set. (TIF)

**S3 Fig. related to Figs 3 and 4 Transfer scores computed using slopes acquired when forcing the regression line intercept zero.** Violin plots show the distributions of transfer scores in A experiment 1 discovery set, B experiment 1 validation set, C experiment 2 discovery set, and D experiment 2 validation set. For A & B (experiment 1), the central black line is the mean of each group, and the vertical bar is the standard error. They closely resemble Fig 3C and 3D in the manuscript, showing that transfer scores largely locate between 0 and 1 and present no significant difference between serial and parallel groups. S3C and S3D Fig (experiment 2). The central green line is the median of each group, and the vertical bar is the interquartile range. They are also highly similar to Fig 4C and 4D in the manuscript, with the interpolation group showing higher transfer scores than the extrapolation group. Insets present predicted slope (grey bar) along with measured slope (olive bar) in the transfer-new trials. Note that for both discovery and validation sets, significant difference was only found in the extrapolation group. $^*p< = .05$; $^{**}p< = .01$; $^{***}p< = .001$; n.s. non-significant. (TIF)

**S4 Fig. related to Fig 6 Simulated transfer scores of linear regression and likelihood-only models.** A (experiment 1 discovery set) & B (experiment 1 validation set) show modelled transfer scores. The central green line is the median of each group, and the vertical bar is the interquartile range. Transfer scores are larger than 1 for both discovery and validation sets. C (experiment 2 discovery set) & D (experiment 2 validation set) show modelled transfer scores and insets compare slopes predicted by Bayesian model (grey bar) along with linearly modelled slopes (olive bar) of the transfer-new trials. Two important features are found: (1) transfer scores of the interpolation group are higher than those of the extrapolation group and (2) transfer scores of the extrapolation group are not significantly different from 0. $^*p< = .05$, $^{**}p< = .01$, $^{***}p< = .001$; n.s. non-significant. (TIF)

**S5 Fig. related to Fig 6 Exemplar modelled data of the experiment 2 discovery set.** A (sampled exemplar size N = 5) & B (sampled exemplar size N = 20) show the distributions of modelled sensory weights and C (N = 5) & D (N = 20) show modelled transfer scores. Insets of C & D show predicted slope (grey bar) along with exemplar modelled slope (olive bar) of the

transfer-new trials. \*\*\*p< = .001; n.s. non-significant.
(TIF)

**S1 Table. Intercept values of each prior/likelihood condition.**
(PDF)

**S2 Table. Participant numbers after the removal of outliers for grouped analyses of each measures including slope, predicted slope in the transfer phase and transfer score.** Numbers in brackets are outliers who were excluded based on the criteria described in the supplementary methods.
(PDF)

## Acknowledgments

We are grateful to Yuanyuan Li and Sara Chen for their help with recruiting and running participants.

## Author Contributions

**Conceptualization:** Chin-Hsuan Sophie Lin, Marta I. Garrido.

**Data curation:** Trang Thuy Do, Lee Unsworth.

**Formal analysis:** Chin-Hsuan Sophie Lin, Trang Thuy Do, Lee Unsworth.

**Funding acquisition:** Chin-Hsuan Sophie Lin, Marta I. Garrido.

**Investigation:** Trang Thuy Do, Lee Unsworth.

**Methodology:** Chin-Hsuan Sophie Lin, Trang Thuy Do, Lee Unsworth.

**Project administration:** Chin-Hsuan Sophie Lin, Marta I. Garrido.

**Supervision:** Chin-Hsuan Sophie Lin, Marta I. Garrido.

**Validation:** Chin-Hsuan Sophie Lin.

**Visualization:** Chin-Hsuan Sophie Lin.

**Writing – original draft:** Chin-Hsuan Sophie Lin, Marta I. Garrido.

**Writing – review & editing:** Chin-Hsuan Sophie Lin, Trang Thuy Do, Lee Unsworth, Marta I. Garrido.

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
