## [Decision Letter · Decision Letter 0]

10 Jul 2023

Dear Dr Lin,

Thank you very much for submitting your manuscript "Are we really Bayesian? Probabilistic inference shows sub-optimal knowledge transfer." for consideration at PLOS Computational Biology.

As with all papers reviewed by the journal, your manuscript was reviewed by members of the editorial board and by several independent reviewers. In light of the reviews (below this email), we would like to invite the resubmission of a significantly-revised version that takes into account the reviewers' comments.

We cannot make any decision about publication until we have seen the revised manuscript and your response to the reviewers' comments. Your revised manuscript is also likely to be sent to reviewers for further evaluation.

Sincerely,

Daniele Marinazzo

Section Editor

PLOS Computational Biology

Daniele Marinazzo

Section Editor

PLOS Computational Biology

Reviewer's Responses to Questions

**Comments to the Authors:**

Reviewer #1: Review of "Are we really Bayesian? Probabilistic inference shows sub-optimal knowledge transfer"

Reviewer: Michael Landy

This paper is one among a few recent attempts to push on the idea that humans make full Bayesian calculations for perceptual estimation by using Mamassian/Maloney's idea of Bayesian transfer. This paper makes a stronger case for a weak, sub-optimal inference computation using a boatload of online data and indices that tailor the hypothesis test to individual observers. It is a significant addition to the literature and worthy of being published. I do have some comments, however none of them are hugely important.

Specifics, mostly by line number:

117: For what it's worth, as far as I know, Konrad got the splatter of dots task, including the unseen coin dropped in the lake/you get to see the bubbles allegory from my paper with Tassinari and Hudson. Note that that paper also asked whether inference was precisely Bayesian, and indicated that it wasn't. We wrote our paper in response to Körding/Wolpert. There paper, cited a gazillion times, claimed that motor responses were Bayesian when, in fact, they FIT a Bayesian observer to the data (rather than measuring the likelihood and training a prior as you and we did).

Later on in the paper, especially in the Discussion, you point out that although Bayesian calculations in general can be quite complex, in a learning design with a fixed prior and likelihood, all Gaussian, then observers needn't do the Bayesian calculation at all and need only learn a weight. I was very happy with this overt discussion of the simplicity of the Gaussian case. That kind of observer doesn't know how to generalize and, without feedback, you'd expect they wouldn't. I like the idea that they might use this learned slope for a variety of conditions and reason toward a guessed slope for new conditions when the new condition is, well, inside the convex hull of what they've learned before (as you talk about in the Discussion). I sign all my reviews, so it's not inappropriate to mention that I've complained to Konrad in multiple reviews that claiming Bayesian when all that's needed is to learn a slope (i.e., do linear regression) is, well, unconvincing. I've also claimed that people seem indiscriminable from optimal in various contexts, but clearly that claim won't always hold!

Minor thought: The slope is constrained in a Bayesian calculation to lie between 0 and 1. First, that might result in range effects (i.e., estimates that avoid the ends of the range), but it's also true that data can result in slopes outside the range. I doubt this bears discussion in the paper, though. On the other hand, you report slopes from a linear regression with an intercept, i.e., you don't force the regression line to go through the center of the prior (i.e., the center of the display). Would doing so change any conclusions? Do you have evidence for intercepts significantly different from zero? (I hope not...). Note: your transfer score is about slopes that come out too low, but if you extrapolate to a prior that is narrow (something you didn't do in your experiment), a conservative (too small) extrapolation of slope would lead to a measured slope that was too big. Your transfer score, however, deals with that outcome correctly, given that you look at change in slope.

148 or so: I was surprised you see to calculate slope here from the entire learning phase, rather than just from later trials. You don't do anything to demonstrate learning here or measure its time course, or show that it has asymptoted by the end of the phase.

Figure 3: You deal with this later with the transfer score, but you never say explicitly that the graphs in this figure pool participants together and thus the predicted slopes (the dashed and dotted lines) are based on the average (or median?) subject, and are not appropriate for individual subjects.

166, 169: Here you are reporting non-significant effects (accepting the null) as evidence for the null. Please don't do that! Later, you do report a Bayesian test (which apparently doesn't show evidence for the null either), but here it's simply a no-no.

I'm not sure why S1 is kept out of the main text. It's the more appropriate analysis than the pooled-participant one in Figure 3.

Fig. 4: Inlets -> Insets

240: "equivalent evidence" isn't a thing, is it? More like equivocal, or simply no evidence either way.

256: What's "Bayesian-fashioned"? ;^)

301: "its complex calculations" are not complex for this Gaussian case

The optimality index is a bit odd. You say it's a trial-by-trial statistic, but you already demonstrate that the likelihood width is wider than simply the dot variance divided by the number of dots. Thus, there are additional sources of noise in observer responses, such as noisy measurement of dot positions, noisy calculation of the centroid, noisy calculation of the weighted average and/or noisy positioning of the net. Noise at the output (motor noise in positioning the net) would affect all likelihood/prior combinations and could affect your conclusions (we discuss an analogous issue with late noise in one of the optimal-cue-integration papers (Jamie Hillis with me, Marty Banks and others, in JOV). It would have been nice if you had a task to estimate the output noise, which is also involved in the likelihood-only task (i.e., that task is not only likelihood!). Also, on a given trial, you know where the centroid was, but you don't know what the internal centroid-location measurement was, so your estimate of optimality index is not, and can't be, based on the internal evidence the participant had to work with.

484: Why the script L in the running text but not in the display equations (for likelihood variance)?

502/Eq. 2: This is subject-specific variance, but is NOT subjective variance (that would be the variance the subject has in his/her head, which could indeed differ from their actual variance).

505: I like the idea of inferring these parameters of the individual subjects for testing transfer. However, the formula here for the variance of the prior is a ratio and a bit complex and likely is typically a biased estimate, which is worth worrying about.

Reviewer #2: The authors present two studies aimed at finding out whether the computations supporting performance in a spatial estimation task are Bayesian. Knowledge transfer was the key criterion used to assess Bayesian computation—to the extent that information about previously learned priors and likelihoods could be used when combined in novel ways (Experiment 1) or when being tested on previously unexperienced likelihoods (Experiment 2), reflects the degree to which computation conforms to Bayesian expectations.

Although performance was qualitatively Bayesian in all cases—combination of prior and likelihood information tended to be sub-optimal. Interestingly, combination of prior and a novel likelihood was better under “interpolation” conditions vs. “extrapolation”. That is, when the novel variability in stimulus information was included within the range of previously experienced stimuli.

Overall, I found the manuscript to be interesting, and addressing an important question in a principled and sophisticated manner. I do have a few questions and concerns that I present below in no particular order.

1 – One of the big issues for me is that the paper is framed in terms of asking whether or not computation is Bayesian, yet the question is addressed entirely within a Bayesian framework. To my mind, the approach taken by the authors cannot address the question of whether computation is Bayesian, but merely the extent to which performance can be approximated by Bayesian computation. This is an important distinction: even if performance is well-characterized by Bayesian computation, it does not follow that this is the underlying process. The same is true for any model-based description of behavior, Bayesian or not. This point is acknowledged by the authors to some extent in the discussion on pp. 19-20, but a more up-front declaration would be good to see.

2 - Related to the above, I was struck by the absence of any comparison with an explicitly heuristic decision process. Although the authors state on page 19 that “There was no evidence supporting the use of a heuristic strategy in the experiment 1,” later on they also mention “However, we cannot fully rule out that participants capitalised on the feature of the coin task to behave in a Bayesian fashion using a special case of heuristics.” It is a little unclear to me how the authors were able to rule out—within limits—the use of heuristic decision strategies. My sense is that doing so would require specification of certain kinds of heuristic strategies and comparing human performance against an appropriately specified model. How would certain heuristic strategies perform in terms of the performance metrics (transfer score and optimality index) the authors presented?

3 – Reflecting on other potential approaches to the task that go beyond purely relying on the prior or likelihood, to what extent might people have been relying on exemplar memory to perform the task? Remembered instances of the true location of the coin, although sampled from the prior, would only gradually approximate the prior over time. This strategy would also intuitively produce an interpolation benefit—stored exemplars would cover the range of locations for interpolation transfer, but not extrapolation transfer.

4 – I was a little confused about the computation of the transfer scores vs. the observed distribution of scores plotted in Figures 3 and 4. The measure is described as normalized, but because it can be positive/negative, so presumably it spans the interval [-1, 1]. I was unsure then how transfer scores greater than 1 were obtained. My guess is that this would reflect a different kind of departure from optimality, overweighting the distribution of stimuli (likelihood) but discounting the prior?

5 – I struggled to follow the description of what was being depicted in Figure 6.

Reviewer #3: The article evaluates human performance in tasks that can be modeled with Bayesian decision theory BDT. The elements of BDT are well known: prior, likelihood, loss function, etc. and whenever a human observer is given correct estimates of the objective prior, etc. we can compare human performance to ideal performance minimizing expected loss.

The authors claim that it has been experimentally established that human performance in Bayesian tasks is close to Bayes optimal but that there is still controversy over whether BDT is an accurate model of human performance. However, this controversy is justified. There is considerable experimental evidence dating to the 1950’s that human observers in signal detection theory (Zhang & Maloney, 2012) distort the prior information given to them a phenomenon known as conservatism. Signal detection theory is a special case of BDT. This observed distortion precludes any claim that humans are Bayesian in all tasks.

The authors should acknowledge that it is established that BDT is approximately correct for some tasks (the ones they cite) and not at all for others (Zhang & Maloney, 20212 reviews many examples of Bayesian failure due to distortion of probability).

The answer to the question posed in the title is “ Depends on the task ”

But the major thrust of the paper is not to evaluate the Bayesian computation but rather to test whether the brain treats different priors as modules that can be extracted from any particular task and transferred to another task. Maloney & Mamassian (2009) referred to this test as a transfer test. In detail: suppose an observers learn a prior and a loss function in one Bayesian task and a different prior and a different loss function in a second task. The learning process may be slow if the tasks are complex. Can the observer then perform a novel task with the prior from the first task and the loss function from the second without further learning?

Maloney & Mamassian point out the work of Trommershauser, Maloney & Landy contains such transfer tasks which the observer successfully negotiates. You might cite Maloney & Mamassian earlier in the article where you introduce their test.

The central part of the article is a series of transfer tests and the experimental work an the results of these tests are very well done. Measuring degree of transfer is elegant. My main criticism of the article is that the conceptual framework is incorrect or misleading but the experimental work reported is of significant value.

**Have the authors made all data and (if applicable) computational code underlying the findings in their manuscript fully available?**

Reviewer #1: Yes

Reviewer #2: **No: **Details will be made available upon acceptance, I presume.

Reviewer #3: Yes

PLOS authors have the option to publish the peer review history of their article (what does this mean?). If published, this will include your full peer review and any attached files.

Reviewer #1: **Yes: **Michael S Landy

Reviewer #2: No

Reviewer #3: No
---

## [Decision Letter · Decision Letter 1]

28 Nov 2023

Dear Dr Lin,

Thank you very much for submitting your manuscript "Are we really Bayesian? Probabilistic inference shows sub-optimal knowledge transfer." for consideration at PLOS Computational Biology. Based on the reviews, we are likely to accept this manuscript for publication, providing that you modify the manuscript according to the review recommendations.

Sincerely,

Daniele Marinazzo

Section Editor

PLOS Computational Biology

Daniele Marinazzo

Section Editor

PLOS Computational Biology

Reviewer's Responses to Questions

**Comments to the Authors:**

Reviewer #1: Re-review of "Are we really Bayesian? Probabilistic inference shows sub-optimal knowledge transer"

Reviewer: Michael Landy

This revision is extremely thorough in response to my and the other reviewers' comments. My only comment is somewhat of a repeat. There's a model used to provide for a predicted subject-specific transfer-phase slope. Now, if you double the prior variance and double the likelihood "variance" (well, the measurement distribution variance), the predicted slope doesn't change. Thus, to get a predicted slope change, you kind of want to nail down one or the other, then learn the other one from condition A, then predict condition B. The authors do this as represented by Eq. 2.

I still find that equation odd as a model of subjective likelihood variance. In the likelihood-only task, the proper thing to do (and hopefully what participants are doing) is point at the centroid (since that's the max-likelihood position). If a participant fails to hit the centroid, the error will be some combination of dot-location-measurement error, centroid-computation error, and response/motor error. This variance will be independent of the error of centroid relative to "true" coin location (that error comes from the independent sampling of the splash). Thus, Eq. 2 calculates the sum of the participant's pointing error variance and the true likelihood variance. Why is this a reasonable way of estimating "subject likelihood variance"? There's nothing subjective in this. Usually, one gets at subjective variance by looking at, e.g., the slope, overtraining the prior, and attributing non-optimal slope as an indication of subjective variance. Thus, in others' papers, the only way to get subjective variance is to assume Bayesian optimal calculation with distorted inputs (e.g., incorrect subjective likelihood variance). So, a large stretch of your paper is based on a model of subjective likelihood variance that, frankly, I don't accept. I'm not sure what to do about that.

Otherwise, all good!

Reviewer #2: I thank the authors for their thoughtful and comprehensive responses to my earlier concerns. Direct comparison of the Bayesian model with the linear regression and exemplar-based heuristic models has substantially improved the strength of the claims made by the authors. The more detailed discussion of exactly what we can glean from this research (e.g., going beyond simply comparing performance against a Bayesian standard; identifying cases of qualitatively Bayesian behavior and diagnosing where processing deviates from optimality) has also improved the clarity of the manuscript. I have no further issues or concerns, and think the manuscript can be accepted in its revised form.

**Have the authors made all data and (if applicable) computational code underlying the findings in their manuscript fully available?**

Reviewer #1: Yes

Reviewer #2: Yes

PLOS authors have the option to publish the peer review history of their article (what does this mean?). If published, this will include your full peer review and any attached files.

Reviewer #1: **Yes: **Michael S Landy

Reviewer #2: No

Figure Files:

Data Requirements:

Reproducibility:

References:

---

## [Editor Report · Decision Letter 2]

18 Dec 2023

Dear Dr Lin,

We are pleased to inform you that your manuscript 'Are we really Bayesian? Probabilistic inference shows sub-optimal knowledge transfer.' has been provisionally accepted for publication in PLOS Computational Biology.

Best regards,

Daniele Marinazzo

Section Editor

PLOS Computational Biology

Daniele Marinazzo

Section Editor

PLOS Computational Biology

---

## [Editor Report · Acceptance letter]

29 Dec 2023

PCOMPBIOL-D-23-00543R2 

Are we really Bayesian? Probabilistic inference shows sub-optimal knowledge transfer.

Dear Dr Lin,

I am pleased to inform you that your manuscript has been formally accepted for publication in PLOS Computational Biology. Your manuscript is now with our production department and you will be notified of the publication date in due course.

With kind regards,

Judit Kozma
